# Trajectories of Seroprevalence and Neutralizing Activity of Antibodies against SARS-CoV-2 in Southern Switzerland between July 2020 and July 2021: An Ongoing, Prospective Population-Based Cohort Study

**DOI:** 10.3390/ijerph20043703

**Published:** 2023-02-19

**Authors:** Rebecca Amati, Giovanni Piumatti, Giovanni Franscella, Peter Buttaroni, Anne-Linda Camerini, Laurie Corna, Sara Levati, Marta Fadda, Maddalena Fiordelli, Anna Maria Annoni, Kleona Bezani, Antonio Amendola, Cristina Fragoso Corti, Serena Sabatini, Marco Kaufmann, Anja Frei, Milo Alan Puhan, Luca Crivelli, Emiliano Albanese

**Affiliations:** 1Institute of Public Health, Faculty of Biomedical Sciences, Università della Svizzera italiana, 6900 Lugano, Switzerland; 2Fondazione Agnelli, 10125 Turin, Italy; 3Faculty of Informatics, Università della Svizzera italiana, 6900 Lugano, Switzerland; 4Department of Business Economics, Health and Social Care, University of Applied Sciences and Arts of Southern Switzerland, 6928 Manno, Switzerland; 5Department of Health Sciences, University of Lucerne, 6002 Lucerne, Switzerland; 6Institute of Microbiology, University of Applied Sciences and Arts of Southern Switzerland, 6501 Bellinzona, Switzerland; 7Epidemiology, Biostatistics and Prevention Institute, University of Zurich, 8001 Zurich, Switzerland

**Keywords:** COVID-19, antibodies, vaccination, cohort study, seroprevalence, acquired immunity

## Abstract

Objectives: The COVID-19 pandemic continues, and evidence on infection- and vaccine-induced immunity is key. We assessed COVID-19 immunity and the neutralizing antibody response to virus variants across age groups in the Swiss population. Study Design: We conducted a cohort study in representative community-dwelling residents aged five years or older in southern Switzerland (total population 353,343), and we collected blood samples in July 2020 (in adults only, N = 646), November–December 2020 (N = 1457), and June–July 2021 (N = 885). Methods: We used a previously validated Luminex assay to measure antibodies targeting the spike (S) and the nucleocapsid (N) proteins of the virus and a high-throughput cell-free neutralization assay optimized for multiple spike protein variants. We calculated seroprevalence with a Bayesian logistic regression model accounting for the population’s sociodemographic structure and the test performance, and we compared the neutralizing activity between vaccinated and convalescent participants across virus variants. Results: The overall seroprevalence was 7.8% (95% CI: 5.4–10.4) by July 2020 and 20.2% (16.4–24.4) by December 2020. By July 2021, the overall seroprevalence increased substantially to 72.5% (69.1–76.4), with the highest estimates of 95.6% (92.8–97.8) among older adults, who developed up to 10.3 more antibodies via vaccination than after infection compared to 3.7 times more in adults. The neutralizing activity was significantly higher for vaccine-induced than infection-induced antibodies for all virus variants (all *p* values < 0.037). Conclusions: Vaccination chiefly contributed to the reduction in immunonaive individuals, particularly those in older age groups. Our findings on the greater neutralizing activity of vaccine-induced antibodies than infection-induced antibodies are greatly informative for future vaccination campaigns.

## 1. Introduction

After the first cases of COVID-19, ascertained in December 2019 in China, severe acute respiratory syndrome coronavirus 2 (SARS-CoV-2) began to spread swiftly around the globe. On 11 March 2020, the World Health Organization (WHO) declared the COVID-19 outbreak a pandemic [1], and, to date, there have been more than 755,385,709 confirmed cases and 6,833,388 deaths globally (10 February 2023, [2]). In spring 2020, the WHO called for regional and national serosurveys, instead of the surveillance of confirmed cases, to estimate the extent of COVID-19 infection in the general population [3].

Several serosurveys have been conducted worldwide to assess the proportion of the population with antibodies against SARS-CoV-2. Marked design, conduction, and quality variations across studies limit comparability and may contribute to the heterogeneity of results and prevent their consolidation [4,5,6,7,8,9,10,11,12]. Furthermore, published serosurveys have a variety of design limitations and methodological drawbacks. For example, data have rarely been collected longitudinally throughout pandemic outbreaks, both before and after the launch of the vaccination campaign [13,14,15,16,17,18]. Moreover, several studies focused on specific sub-populations (e.g., healthcare workers [19,20], children [21,22], or blood donors [23]).

A large corpus of evidence suggests that vaccination has contributed substantially to the increased seropositivity and high neutralizing activity of SARS-CoV-2 antibodies in the general population [24,25,26]. Nonetheless, before the spread of Omicron in 2022, the antibodies’ neutralizing capacity and the difference between infection- and vaccine-induced immunity were rarely accounted for in serosurveys [27,28]. Finally, especially in the first months of the pandemic, the immunoassays used to detect anti-SARS-CoV-2 antibodies were neither designed nor validated for population-based studies, and their accuracy and validity were suboptimal [29,30,31].

We used data from “Corona Immunitas Ticino”, an ongoing population-based study conducted in representative samples of the Swiss population aged five years and above. Here, we focus on southern Switzerland (Ticino), a region bordering northern Italy, the European epicenter of the pandemic in 2020 [32,33]. We conducted longitudinal serosurveys, with the baseline in July 2020 and follow-ups in November–December 2020 and six months after the beginning of the vaccination campaign in the region in June–July 2021 [34]. We aimed to describe the temporal and regional variations in the seroprevalence of anti-SARS-CoV-2 antibodies by sex and across age groups, using a previously validated Luminex assay purposely developed for population-based serosurveys [35]. Next, we studied infection- and vaccine-induced immunity, and we compared the proportion of neutralizing antibodies for different SARS-CoV-2 variants in vaccinated and non-vaccinated seropositive individuals as a proxy for the quality of acquired immunity.

## 2. Materials and Methods

### 2.1. Study Design and Participants

“Corona Immunitas Ticino” is a population-based cohort study of a representative sample of community-dwelling residents of Ticino (southern Switzerland) aged five years and older. The study is part of the Corona Immunitas national research program designed to measure the spread of COVID-19 infections and the impact of the pandemic on the general population [36]. Between July and November 2020, we sent 13,931 invitation letters to a sex- and age-stratified random sample drawn from the household registry of Ticino residents, out of a total regional population of 353,343 (as of 31 December 2019).

### 2.2. Measurements and Data Collection

We collected data using the survey function in Research Electronic Data Capture (REDCap), a secure, web-based platform designed to support data capture, collection, and integration [37,38]. Starting from July 2020 (age group of 20–64 years) and September 2020 (age groups of 5–13 years, 14–19 years, and 65+ years), participants registered online or over the phone as preferred, and they reported their baseline sociodemographic, economic, health, and lifestyle information. We enquired about COVID-19 symptoms and vaccination status monthly. Parents filled out the questionnaires with their child aged 5–13 years. Additionally, we trained nine interviewers to collect data from older adults with limited internet access or low digital literacy using a bespoke Computer-Assisted Telephone Interview (CATI) system.

### 2.3. Serological Testing

To measure seropositivity for SARS-CoV-2, we collected peripheral venous blood samples at three time points scheduled after the first, second, and third epidemic waves of the pandemic in Switzerland, i.e., in the second half of July 2020, mid-November to early December 2020, and June to early July 2021, respectively. Hence, the average follow-up times ranged between four and eight months. The participants in all age groups freely chose one of the 17 sites across Ticino and the date and time of their appointments using an online reservation system, and they provided contextual information about their COVID-19-related symptoms and infection exposure using a dedicated serospecific questionnaire. In July 2020, we tested only non-vulnerable adults aged 20–64. In November–December 2020, we included children, adolescents, and older adults, and we offered optional home visits for blood withdrawals to vulnerable individuals. We used a previously validated Luminex assay for anti-SARS-CoV-2 total immunoglobulins, purposely developed for population-based serosurveys [35]. In June–July 2021, given the ongoing vaccination campaign from January 2021, we measured antibodies targeting the spike (S) and nucleocapsid (N) proteins of the virus. While the latter only develop following natural infection, S antibodies can develop following natural infection, as well as after vaccination, facilitating the distinction between infection- and/or vaccine-induced immunity (below) [35]. Next, we used a previously validated high-throughput and cell-free neutralization assay that allows for the simultaneous evaluation of multiple spike protein variants [35,39], including Alpha (Phylogenetic Assignment of Named Global Outbreak, Pango lineage designation B.1.1.7), Beta (Pango lineage B.1.351), Gamma (Pango lineage P.1), and Delta (Pango lineage B.1.617.2), to measure the neutralizing activity of antibodies (i.e., serum dilution IC_50_ greater than 50) in a random sub-sample of participants 14 years and older who were seropositive at the third serosurvey in June–July 2021 (N = 250). Finally, we compared the neutralizing activity between the vaccinated and convalescent participants by age group.

### 2.4. Statistical Analysis

We assumed a minimum seroprevalence of 7% based on regional data on cumulative, laboratory-confirmed COVID-19 cases after the first pandemic wave in Ticino [40], and we calculated sample sizes to allow for adequate precision in the estimation of SARS-CoV-2 seroprevalence with 80% power, accounting for non-participation and attrition. We calculated the seroprevalence estimates with 95% CI using a Bayesian logistic regression model accounting for the target population’s sociodemographic structure and for the measurement properties of the Luminex test [41]. We formally compared the proportions of seropositives by age, between serosurveys, and by district with standard Pearson’s Chi2 tests.

The vaccination campaign began in southern Switzerland on 4 January 2021. We computed the proportion of participants who received two doses of the vaccine by the beginning of July 2021 by age group, and we combined the information on vaccination status with the serological results to adjudicate infection- and vaccine-acquired immunity. The groups were defined as follows: (1) Seronegatives = anti-SARS-CoV-2 antibodies not detected. (2) Infection-induced seropositives = both anti-S and anti-N antibodies positive irrespective of self-reported vaccination status or only anti-S antibodies positive in those who reported that they were not vaccinated. Anti-N antibodies wane more quickly than anti-S antibodies. If a participant only had anti-S antibodies but reported not to have been vaccinated, we assumed that their antibodies had developed following an infection, irrespective of the negativity of the anti-nucleocapsid antibodies. (3) Vaccine-induced seropositives = anti-S antibodies positive, anti-N antibodies negative, and self-reported vaccination.

### 2.5. Ethics

All participants signed paper versions of informed consent at each blood-sampling follow-up. The Ethics Committee of Ticino (part of SwissEthics) authorized the study (BASEC- 2020-01514) on 23 June 2020. 

## 3. Results

We enrolled 3028 participants in the digital cohort (629 children 5–13 years, 451 adolescents 14–19 years, 1049 adults 20–64 years, and 882 older adults 65+). For the three serosurveys, we sampled participants aged five years or more from the 3028 of the ongoing cohort irrespective of seropositivity at the previous follow-ups. In July 2020, after the first pandemic wave, we successfully performed 646 serological tests (74% response rate) in adults aged 20–64 years. From mid-November to early December 2020, after the second pandemic wave, we performed 1457 serological tests (70% response rate). In June–July 2021, after the third pandemic wave, we completed 885 serological tests (45% response rate) for the third serosurvey (Figure 1).

### Descriptive Data

Of the 646 adults (20–64 years) who took part in the first serosurvey, 57.4% were females. At the second serosurvey, 54.3% were females, and we tested 349 children aged 5–13 years, 284 adolescents aged 14–19 years, 210 adults aged 20–64 years, and 614 adults 65+. At the third serosurvey, 51.5% were females, and we tested 166 children, 101 adolescents, 300 adults, and 318 older adults (Table 1 and Figure 1). Compared to the age structure of the target population of southern Switzerland, in the study sample children, adolescents and older adults were slightly over-represented, and adults were under-represented. Our study sample can be considered fairly representative of the whole population [42].

Overall, educational attainment was high in all age groups, 66.2% of adults were employed, and 94.6% of older adults were retired. The number of people per household was either one or two in nearly half of the sample (48.1%).

The seroprevalence in July 2020 was 7.8% (95% CI: 5.4–10·4), estimated in 646 adults aged 20–64 years. By December 2020, the seroprevalence had tripled to 26.3% (95% CI: 20.4–32.9) in adults, and it was 14.8% (95% CI: 11.1–19.0) in children, 17.7% (95% CI: 13.3–22.9) in adolescents, and 7.2% (95% CI: 4.6–9.9) in older adults. The proportion of the study sample that developed antibodies against the virus increased sharply in all age groups by July 2021, reflecting the launch (January 4, 2021) and advancement of the COVID-19 vaccination campaign: It more than doubled among children (35.1%, 95% CI: 28.2–42.4), for whom the vaccine was not yet available, and among adolescents (46.5%, 95% CI: 36.5–56.6), for whom the campaign had just started. It more than tripled among adults (71.4%, 95% CI: 66.1–76.7), and it skyrocketed to 95·6% (95% CI: 92.8–97.8) among older adults, who had complete access to the vaccine (Table 1).

At the second and third serosurveys, the seroprevalence significantly differed by age (Chi^2^ > 55 with 6 df, all *p* < 0.001), and the temporal increases over all the serosurveys are in line with the cumulative incidence of laboratory-confirmed cases according to the FOPH statistics [40] (Figure 2).

The proportion of still immunonaive older adults (i.e., those who had not developed antibodies) was the highest in December 2020 and the lowest in July 2021, and it was inversely associated with age for the remaining age groups at the last follow-up, when the vaccines were not yet approved for children. All the observed increases over time in seroprevalence were statistically significant within age groups (Chi^2^ > 100.1 with 4 df, and all *p* values < 0.001). Estimates varied by up to 10% within the regional districts of southern Switzerland, but seroprevalence was not significantly higher in the southern districts bordering the Italian region Lombardy (*p* > 0.05 for all comparisons), which was heavily hit during the first pandemic waves (Figure 3) [33].

The COVID-19 vaccination campaign in southern Switzerland began with vulnerable groups and individuals older than 85 years on 4 January 2021, and it expanded eligibility by 5-year age intervals every other week [34]. In June–July 2021, when we conducted the third serosurvey, the vaccination campaign had just started in adolescents (12–15 years) [43] but not yet in children (younger than 12 years), for whom it was launched in January 2022 [44]. A total of 63.6% of participants in our study reported having received at least one dose of an anti-COVID-19 vaccine (0% of children, 30.7% of adolescents, 68% of adults, and 94.3% of older adults). At the time of blood sampling, 528 individuals (59.7% of the actual study sample of N = 885, reported in Table 1) had only anti-spike antibodies. Seroconversion was not complete for 52 (19.6%) of the 266 who were still seronegative but had already received one dose of a vaccine (see the group “seronegatives” in Table 2 and Figure 4). There were 29 individuals who were both infected and vaccinated.

Age was inversely associated with seronegativity and positively associated with vaccine-induced seropositivity. By July 2021, for each person with infection-induced anti-SARS-CoV-2 antibodies, there were nearly four people with vaccine-induced antibodies in the study sample. This ratio varied between 0 in children (none of whom were vaccinated), 1.1 in adolescents, 3.7 in adults, and 10.3 in older adults (Table 2 and Figure 4).

The proportion of participants whose antibodies demonstrated a serum dilution IC_50_ greater than 50 (i.e., above-threshold neutralizing activity) varied by SARS-CoV-2 variant (Table 3 and Figure 5 and Figure 6).

For the whole sample of vaccinated and non-vaccinated individuals, we found that the neutralizing activity against SARS-CoV-2 variants was higher for the spike wild type (93%) and progressively lower for the Alpha (89%), Gamma (85%), Delta (78%), and Beta (77%) variants. These proportions also varied by age: The neutralizing antibodies for the spike wild type were 98% among older adults (65+ years; N = 125), 90% among adults (20–64 years; N = 105), and 75% among adolescents (14–19 years; N = 20). For the Alpha variant, the neutralizing antibodies were 95%, 87%, and 75% in each age group, respectively; for the Gamma variant, they were 92%, 80%, and 65%, respectively; for the Delta variant, they were 90%, 70%, and 50%, respectively; and for the Beta variant, they were 92%, 80%, and 65%, respectively. Neutralizing activity was significantly higher for vaccine-induced than infection-induced antibodies in both adolescents and adults and for all variants (all *p* values for the difference in proportions between groups < 0.037). All older adults in this subgroup were vaccinated, which impeded between-group comparisons (Figure 6).

## 4. Discussion

Between spring 2020 and summer 2021, we designed and conducted a prospective, population-based cohort study to investigate the spread of COVID-19 infections through the first three pandemic waves in southern Switzerland (Ticino). We evaluated the contribution of vaccination to the reduction in immunonaive individuals, and we measured the neutralizing activity of circulating antibodies against five common variants of SARS-CoV-2.

According to our findings, the proportion of the population with anti-SARS-CoV-2 antibodies was below 10% after the first wave in July 2020; it doubled by December 2020 and skyrocketed by July 2021, with great variations by age. For each person with infection-induced anti-SARS-CoV-2 antibodies, there were nearly four adults aged 20–64 and ten older adults with vaccine-induced antibodies by July 2021. The neutralizing activity of the circulating antibodies was lower for more recent variants of the virus, including Delta, and it was significantly higher in vaccinated than in non-vaccinated individuals.

The estimated seroprevalence in southern Switzerland in the first, second, and third serosurveys is consistent with that reported in Geneva (western Switzerland) after the first [41], second [45], and third pandemic waves [46] and that in the canton of Zurich (northern Switzerland) in children [47], but it is markedly higher in summer 2021 (72.5% vs. 22.3%) than the results of a cohort study conducted by local health authorities in Ticino [48], owing to differences in the serological test. While our test was validated in and optimized for population-based samples and facilitated the detection of antibodies developed after vaccination and/or after infection, local health authorities used a rapid test that detected only anti-nucleocapsid antibodies, which wane within a few months [49,50,51] and do not develop following vaccination.

We found progressive temporal increases but fairly stable regional differences in seroprevalence across the pandemic waves. Comparisons are not straightforward with other international studies that reported marked regional variations in seroprevalence after the first and second pandemic waves [17,18,45,52,53]. These variations may be attributable, at least in part, to differences in study design and measurement features, but also to the period of data collection. For example, the REACT-2 study in the UK was based on finger-prick antibody tests, which are cheaper, more practical, and less invasive but also less accurate and more prone to measurement bias and errors than laboratory-based antibody assays [54]. Home testing may also introduce self-selection bias and contribute to an over-estimation of true prevalence. Moreover, the fifth round of the REACT-2 study was completed two months after the launch of the UK vaccination campaign. We completed our third round more than six months after the commencement of the stepwise vaccination campaign in Switzerland, and we used a high-quality serological testing method of the acquired immune response. We found increases in seroprevalence over time that are consistent in magnitude with evidence from studies that used repeated cross-sectional designs [9,55,56]. Our findings on the marked variation in the proportion of seroprevalence due to vaccination coverage are based on data collected before the spread of both the Omicron variant and the ensuing rapid increases in hybrid immunity in populations (whereby both vaccines and infection induced antibodies). There may be relevant implications for infection prevention in population sub-groups because it is plausible that increases in seroprevalence differed by age because of both varying risk exposure and age-varying susceptibility to infection due to circulating variants.

Our findings suggest that the immune response to COVID-19 vaccines in older adults was adequate after two doses and endured for six months from the first dose. Although further boosters may be needed [57], our results provide epidemiological support for the experimentally demonstrated efficacy of the available vaccines [24,25,58] and for their immunogenicity for the virus variants investigated [59] (as illustrated in Figure 6). Furthermore, our observations on the inverse association of the proportion of samples that could efficiently neutralize spike proteins with increasing viral mutations after two doses of a vaccine and before the circulation of the Omicron (B.1.1.529) variant extend preliminary findings obtained in mechanistic studies [35] and in selected clinical samples [60,61]. Similar to findings recently reported [19,21,24,25,62,63,64,65], neutralizing antibody titers were higher in the vaccinated than in the non-vaccinated individuals.

Our findings on constant, steep, and age-varying increases in seroprevalence through the pandemic waves, between summer 2020 and summer 2021, are in line with similar increases in regional data on cumulative, laboratory-confirmed COVID-19 cases [40]. However, compared to monitoring data, our estimates provide a more accurate account of the actual spread of the infection in the population and of the varying proportions of immunonaive individuals across age groups, irrespective of test accessibility, availability, and help-seeking behaviors, and they account for the contribution of vaccination to acquired immunity.

In December 2020, the great majority of older adults were still immunonaive. The course of the disease is usually more severe [66], and infection fatality rates were the highest in this age group before vaccination [67,68,69]. However, although vaccines protect against severe disease outcomes, preliminary evidence suggests that the waning of vaccine-induced antibodies is more rapid in older than in younger adults [67,70]. The data collection in our study was completed months before the rapid spread of the Omicron variant. Our results clearly indicate that vaccination was the major contributor to the observed increase in seroprevalence. In older adults, vaccination coverage and vaccine-induced immunity were both around 95% by July 2021. Our findings on the key contribution of vaccine-induced immunity in older adults have implications for vaccine boosters in this age group. Clinicians may advise their patients to get vaccinated, and they may consider using serological testing that distinguishes between infection- and vaccine-induced immune responses. Because vaccinated individuals acquire good functional immunity but remain potentially infectious and hybrid immunity may confer higher immune protection than vaccine- or infection-induced immunity alone [19,21,63,64,65], serosurveys remain key to assessing the dynamics of antibody waning and vaccine breakthrough infections in the population [71].

Our results on the potential reduction in neutralizing levels against newer variants of the virus (with the apparent exception of Beta) may have important public health implications. Vaccines confer very high individual protection against COVID-19 symptoms [72], hospitalization [73], and death [74] in the months after injection, and they can substantially contribute to reducing both the mortality associated with COVID-19 and the pressure on health services [75,76,77,78,79]. Although protection against severe re-infection may decrease as new variants emerge, our findings on the higher neutralizing levels against all SARS-CoV-2 variants in those with vaccine-induced immunity than in those with infection-induced immunity are in line with evidence on the reduced COVID-19 mortality associated with booster doses of vaccines [80], and they provide support to continue, adapt, and strengthen vaccination campaigns in all age groups and to plan for booster doses.

Some limitations are worth noting. First, the response rates decreased over time. The external validity of our findings may vary accordingly. Second, lower and higher risk exposure could have negatively and positively influenced participation, respectively. However, the sociodemographic characteristics of the actual study sample are comparable to those of the target population and do not differ from those of non-respondents [42]. Third, we used a serological test that required peripheral blood drawing, which is more invasive and potentially riskier than finger-prick antibody tests. This may have influenced participation but plausibly non-differentially with respect to infection status. Selection bias seems unlikely. Moreover, we offered proximity blood-sampling options and home-based visits to vulnerable individuals, including older adults. Fourth, it is possible that the striking increases in seroprevalence in the population modified the positive predictive value of the serological test [35]. However, the fact that the accuracy and validity of our assay were demonstrated in population-based samples is a major strength of our study [81]. Moreover, we accounted for the sensitivity and specificity of the test, and we used a robust Bayesian model to adjust the estimates for both false positives and false negatives. Our estimates are valid and reliable, and they can be generalized with confidence to similar and neighboring populations. The higher participation of older adults, the large majority of whom remained immunonaive throughout the second pandemic wave, may have led to an underestimation of overall seroprevalence in winter 2020. Fifth, we combined self-reported information on vaccination status with serological results to distinguish the infection-induced seroprevalence of antibodies from the vaccine-induced seroprevalence of antibodies. Neutralizing activity may be somewhat boosted in the latter group and may have inflated our results on the positive association between vaccination status and neutralizing activity, and some overlaps between categories may exist because anti-N antibodies wane faster than anti-S antibodies [49,51,81]. Finally, we cannot exclude that, to some extent, a lower neutralization may also be due to a longer time since the induction of antibodies after infection compared to vaccination. Longer follow-ups may be needed to shed light on this and other pending uncertainties.

## 5. Conclusions

In conclusion, good-quality serosurveys remain indispensable to determine the actual extent of COVID-19 infections in populations, and immunoassays provide crucial insights into the acquired immune response in community settings, including hybrid immunity and antibodies’ neutralizing capacity, as new variants emerge and vaccination campaigns continue. Our study provides accurate estimates of changes in the proportion of immunonaive individuals in southern Switzerland through the pandemic waves, and it highlights marked differences across age groups. Our results have significant public health implications at the local, national, and international levels because increases in seropositivity until 2021 were not due to infections but primarily to vaccination. Further investigations in population-based samples on antibody neutralizing activity by vaccination status are warranted, particularly if and after new virus variants emerge.

## Figures and Tables

**Figure 1 ijerph-20-03703-f001:**
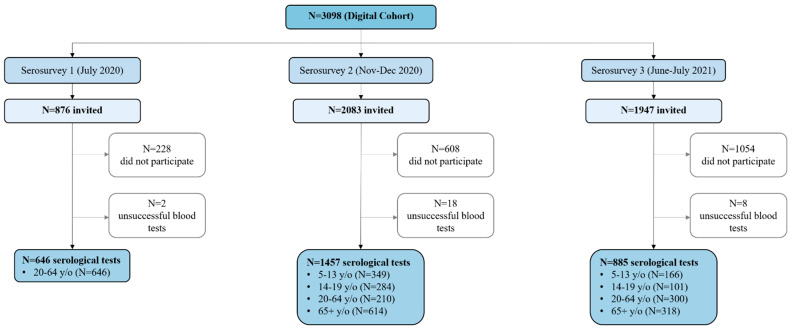
Flow diagram of participants at each serosurvey of the “Corona Immunitas Ticino” study.

**Figure 2 ijerph-20-03703-f002:**
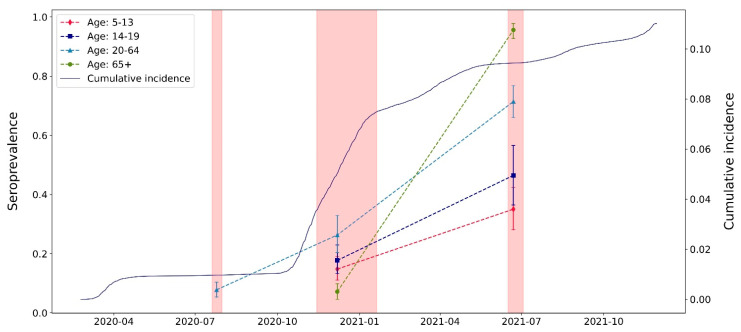
Seroprevalence estimates with 95% CI at each serosurvey (vertical red banners), and cumulative incidence rate of laboratory-confirmed COVID-19 cases in the region (solid line).

**Figure 3 ijerph-20-03703-f003:**
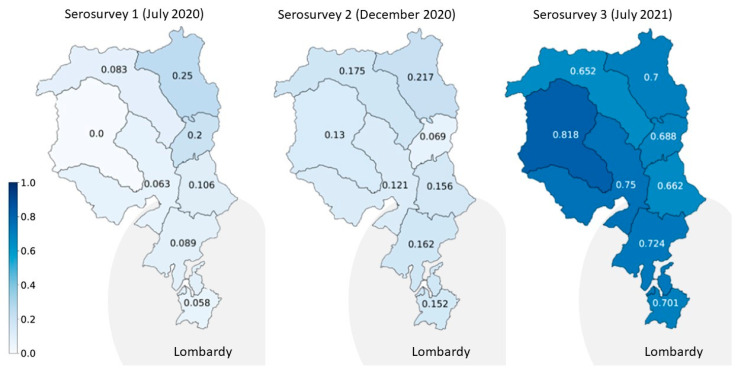
Temporal evolution of proportion of seropositive participants in different geographic regions of southern Switzerland (Ticino). The Italian Lombardy region borders the southern part of Ticino.

**Figure 4 ijerph-20-03703-f004:**
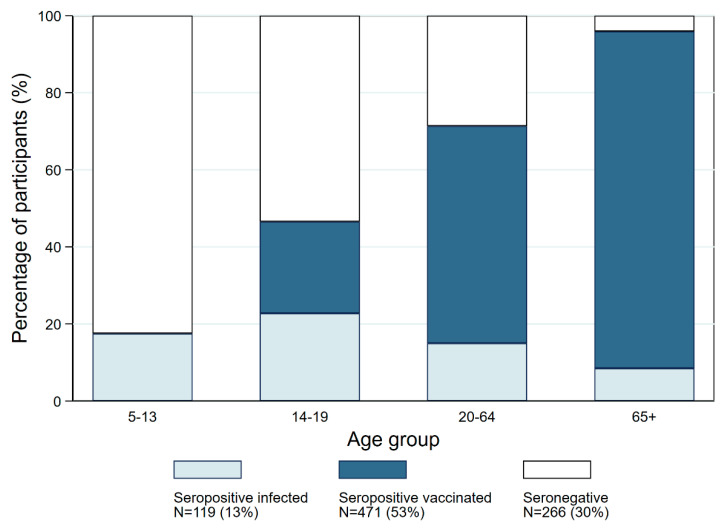
Rates of infection- and vaccine-induced immunity by age group at the third serosurvey in June 2021 (N = 885). Seronegatives = anti-SARS-CoV-2 antibodies not detected; infection-induced seropositives = both anti-S and anti-N antibodies positive irrespective of self-reported vaccination status or only anti-S antibodies positive in those who reported that they were not vaccinated; vaccine-induced seropositives = anti-S antibodies positive, anti-N antibodies negative, and self-reported vaccination. Individuals who were infected and vaccinated (S+ N+ vaccinated) (N = 29) are not reported.

**Figure 5 ijerph-20-03703-f005:**
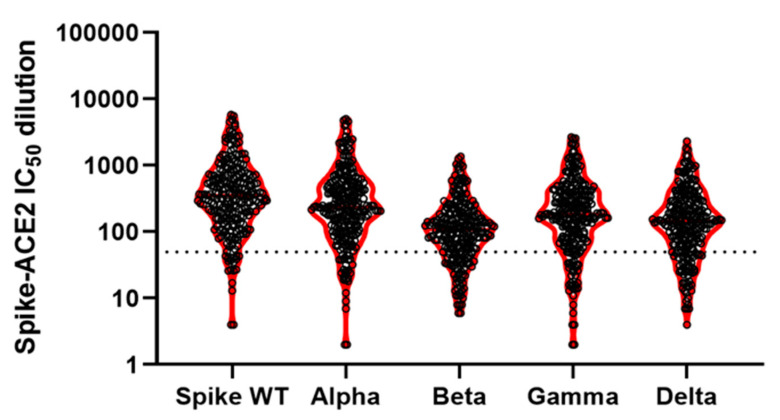
SARS-CoV-2 antibodies’ neutralizing activity by variant in the sub-sample of seropositive participants at the third serosurvey in June–July 2021 (N = 250).

**Figure 6 ijerph-20-03703-f006:**
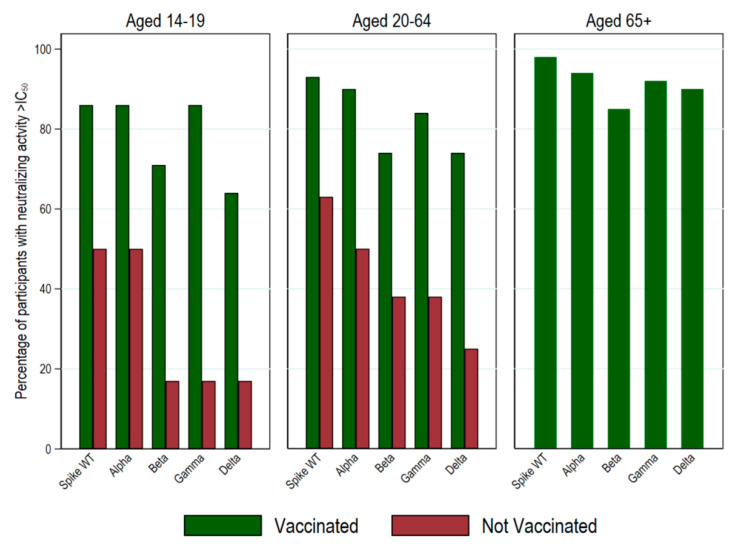
Neutralizing activity against SARS-CoV-2 variants by vaccination status across age groups in the sub-sample of seropositive participants at the third serosurvey in June–July 2021 (N = 250: 65+ years N = 125; 20–64 years N = 105; 14–19 years N = 20). The *y*-axis indicates the percentage of participants with an above IC_50_ threshold neutralizing activity by self-reported vaccination status at the June–July 2021 serosurvey.

**Table 1 ijerph-20-03703-t001:** Seroprevalence estimates through the COVID-19 pandemic waves in southern Switzerland.

	Serosurvey 1 (July 2020)		Serosurvey 2 (November–December 2020)		Serosurvey 3 (June–July 2021)	
	Number Tested (Positives)	Seroprevalence (95% CI)	Number Tested (Positives)	Seroprevalence (95% CI)	Number Tested (Positives)	Seroprevalence (95% CI)
Overall	646 (54)	7.8% (5.4–10.4)	1457 (211)	20.2% (16.4–24.4)	885 (618)	72.5% (69.1–76.4)
Age group						
5–13	-	-	349 (53)	14.8% (11.0–19.0)	166 (53)	35.1% (28.2–42.4)
14–19	-	-	284 (52)	17.7% (13.3–22.9)	101 (46)	46.5% (36.5–56.6)
20–64	646 (54)	7.8% (5.4–10.4)	210 (56)	26.3% (20.4–33.7)	300 (214)	71.4% (66.1–76.7)
65+	-	-	614 (50)	7.2% (4.6–9.9)	318 (305)	95.6% (92.8–97.8)
Sex						
Female	371 (30)	7.9% (5.1–11.1)	787 (104)	17.8% (14.7–22.3)	452 (307)	70.5% (65.9–74.9)
Male	275 (24)	7.7% (4.5–11.4)	670 (107)	22.7% (18.1–28.6)	433 (311)	74.5% (70.1–78.8)

Note: We calculated seroprevalence estimates with 95% CI using a Bayesian logistic regression model accounting for the target population’s sociodemographic structure and for the measurement properties of the Luminex test.

**Table 2 ijerph-20-03703-t002:** Seronegative, infection-induced seropositive, and vaccine-induced seropositive participants for SARS-CoV-2 antibodies by age group at the third follow-up (June–July 2021), expressed as numbers (percentages), if not otherwise specified.

	Age Group				
	5–13	14–19	20–64	65+	Total
Seronegatives	113 (82.5)	54 (53.5)	86 (28.7)	13 (4.1)	266 (31.1)
Infection-induced seropositives	24 (17.5)	23 (22.8)	45 (15.0)	27 (8.5)	119 (13.9)
Vaccine-induced seropositives	0 (0)	24 (23.8)	169 (56.3)	278 (87.4)	471 (55.1)
Ratio of infection- to vaccine-induced seropositivity	0.00	1.1	3.7	10.3	3.9
TOTAL	137	101	300	318	856

Note: seronegatives = anti-SARS-CoV-2 antibodies not detected; infection-induced seropositives = both anti-S and anti-N antibodies positive irrespective of self-reported vaccination status or only anti-S antibodies positive in those who reported that they were not vaccinated; vaccine-induced seropositives = anti-S antibodies positive, anti-N antibodies negative, and self-reported vaccination. Total sample sizes in Table 1 and Table 2 differ because individuals who were infected and vaccinated (S+ N+ vaccinated) (N = 29) are not reported in the table.

**Table 3 ijerph-20-03703-t003:** Presence of SARS-CoV-2 antibodies and neutralizing activity by variant in a random sub-sample of seropositive participants at the third serosurvey in June–July 2021 (N = 250).

		Anti-Spike Antibodies ^a^	Anti-NuC Antibodies ^a^	Spike Wild Type ^b^	Alpha ^b^	Beta ^b^	Gamma ^b^	Delta ^b^
All	Mean:	85	2	651	500	172	339	247
	N positive:	250	29	232	223	192	212	196
	% positive:	100	12	93	89	77	85	78
	N total:	250	250	250	250	250	250	250

**^a^** The levels of anti-spike and anti-nucleocapsid IgG antibodies were assessed with the Mean Fluorescence Intensity (MFI) ratio, as measured using the Luminex binding assay Sensitive Anti-SARS-CoV-2 Spike Trimer Immunoglobulin Serological (SenASTrIS). The threshold of positivity is ≥6 for both anti-spike and anti-NuC IgG antibodies [39]. In the first two columns, we report the overall mean of the levels of anti-spike and anti-NuC IgG antibodies of all participants in the sample (N = 250) and the number (N positive) and percentage (% positive) of participants with a mean value above the ≥6 threshold. **^b^** For the wild type and each variant, we report the neutralizing activity of antibodies expressed as mean values of spike-ACE2 IC_50_ dilution (calculated based on the overall sample, N = 250) and the number (N positive) and percentage (% positive) of participants with a mean value above the IC_50_ threshold for neutralizing activity.

## Data Availability

Deidentified individual participant data underlying the findings of this study will be available for researchers submitting a methodologically sound proposal to achieve the aims of the proposal after the publication of this article. Proposals should be directed to the corresponding author (Emiliano Albanese, emiliano.albanese@usi.ch).

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
