# Peer review of "Trajectories of Seroprevalence and Neutralizing Activity of Antibodies against SARS-CoV-2 in Southern Switzerland between July 2020 and July 2021: An Ongoing, Prospective Population-Based Cohort Study"

_ijerph, 2023, doi:10.3390/ijerph20043703_

Round 1

Reviewer 1 Report

The authors calculated seroprevalence with a Bayesian logistic regression model accounting for the population’s sociodemographic structure and the test performance, and compared the neutralizing activity between vaccinated and convalescent participants, across virus variants. Results. Overall seroprevalence was 7.8% (95% CI: 5.4-10.4) by July 2020, and 20.2% (16.4-24.4) by December 2020. By July 2021, overall seroprevalence increased substantially to 72.5% (69.1-76.4) with highest estimates among older adults of 95.6% (92.8-97.8), who developed antibodies via vaccination up to 10.3 more than after infection compared to 3.7 times more in adults. The neutralizing activity was significantly higher for vaccine-induced compared to infection-induced antibodies for all virus variants (all p values < 0.037).  Vaccination drastically contributed to the reduction of immunonaive individuals, particularly in older age groups. Our findings on the greater neutralizing activity of vaccine- compared to infection-induced antibodies are greatly informative for future vaccination campaigns.

(PUT INSERTS IN A DIFFERENT COLOR FONT TO IDENTIFY CHANGES IN THE ARTICLE)

Include in article

1 - Abstract

Conclusions: State only what your study found; do not include extraneous information not backed up by the results.

2 - Discussion

Compare and contrast your study with others in the most relevant world literature, particularly the recent literature.

3 - What new information is sufficient to modify existing clinical practice?

4 -What are the conclusions and implications for current practice, and particularly for future research that may have a significant impact on clinical decisions?

5 - How can this study affect public policies related to health?

6 - What does this study add to the literature?

7 - At the end of the Discussion, under the subheading "Limitations," review the limitations of your study.

8 - At the end of the limitations, under the subheading " Future directions".

9 - Conclusion

Take special care to draw your conclusions only from your results and verify that your conclusions are firmly supported by your data

10 - Table 1

I suggest include: BMI, waist circumference, Waist Hip Ratio,  metabolic syndrome, stroke, lipids, lipids ratio, exercise, hypertension, diabetes,  smokers, liver steatosis, alcohol intake, Non-HDL cholesterol, COPD¸ Gout, Carotid artery disease, Hypothyroidism, Anemias, Drug abuse, asthma, atrial fibrillation, high-sensitivity C-reactive protein, Chronic kidney disease, Dialysis, Heart failure, Liver cirrhosis, mental disorders, Gastrointestinal bleeding history, Fasting glucose, Depression, Valve disorder,  Blood pressure (Systolic and Diastolic), Heart rate, bpm; Hemoglobin, g/dL, mean (SD), Platelet, x 109/L, mean (SD)  and medications. Family history of stroke,   CAD, PAD. History of cancer. HIV Status: Positive, n (%)

11– References

Update

12- Supplementary material

A -Include the database used to perform the analysis

B -Include the algorithms and scripts used in the software to perform the statistical analysis.

Author Response

We thank this Reviewer for the comments. Please see the attachment with our point-by-point response.

Reviewer 2 Report

The topic of the paper „Trajectories of seroprevalence and neutralizing activity of antibodies to SARS-CoV-2 in southern Switzerland between July 2020 and July 2021: an ongoing, prospective population-based cohort study” is very interesting for readers because the COVID-19 pandemic endures and evidence on infection- and vaccine-induced immunity being the key.

The aim of the present study was to describe temporal and regional variations in seroprevalence of anti-SARS-CoV-2 antibodies by sex and across age groups, using a previously validated Luminex assay purposely developed for population-based serosurveys. They studied infection- and vaccine-induced immunity and compared the proportion of neutralizing antibodies for different SARS-CoV-2 variants in vaccinated and non-vaccinated seropositive individuals, as a proxy of the quality of the acquired immunity.

The authors concluded that their study provides accurate estimates of changes in the proportion of immunonaive individuals in southern Switzerland through the pandemic waves, and highlights marked differences across age groups. So, further investigations in population-based samples on antibody neutralizing activity by vaccination status are warranted, particularly if and after new virus variants emerge.

The introduction provides sufficient background and includes relevant references.

The manuscript is well written, and the text is easy to read.

The design research is well described.

The results are consistent and clearly presented.

The reference list is variously and relatively recently.

Author Response

We thank this Reviewer for the appreciation of our work.

Reviewer 3 Report

COVID-19 pandemics is the major public health concerns worldwide. Amati and colleagues carried out a population-based study to monitor the seroprevalence in Ticino. They validated the neutralizing activity of SARS-CoV-2 antibodies from sampled serum to different various SARS-CoV-2 variants, and further correlated the antibody level in different age groups to the implication of vaccine campaign. This study is important for future vaccine campaign, but a number of major issues must be addressed.

Major comments:

1.     The resolution of the figures is too low and the labels in the figures are not clear, especially for Fig 1,2 and 6.

2.     As mentioned in the introduction, an assay using N protein antibody is developed to indicate natural infection (line122-123). The definition of infection-induced seropositive is confusing: “infection-induced seropositives  = both anti-S and anti-N antibodies positive irrespective of self-reported vaccination status OR only anti-S antibodies positive who reported they had no vaccination” (line 149, 235, 242) . Further clarification is required.

3.     Line 173-175: the idea is not clear. More precise description is required.

4.     Line 183-193: it is recommended to include the date of the vaccine campaign launched and its relationship with the changes in the seroprevalence.

5.     Line 207-208: how was the significant difference between different regional districts calculated? The southern and western districts bordering the Italian region Lombardy needs to be labeled in Fig 3. Reference is required for the last statement in line 208.

6.     Line 217-220: Reference is required for the mentioned numbers.

7.     Line 220-223: Where does this result come from, especially for those 528 individuals? All the numbers mentioned was not found in figures or tables.

8.     Line 226-228: the idea is not clear. More precise description is required.

9.     Line 231: What does “N(%)” mean?

10.  The number of the infected and vaccinated individuals has to be mentioned in the main text. By looking at Table 2 alone, the total number of participants does not match the total number found in Table 1.

11.  Fig 5: “Spike WT” is recommended to be labeled as “ancestral spike”. What are the criteria to choose those participants (N=250)? Why is the number of participants limited to “250”?

12.  Table 3: More precise description is required to describe the numbers. For example, what do “spike ratio” and “NuC ratio” mean?

13.  Fig 6: The y-axis is not precise, which represents the relative value but not the percentages of participants as mentioned in the figure legend. What are the total participants in each age group? In age group 65+, the figure indicated that all the elderly got the vaccine, but this result is contradicted to the result in Table 1 and Fig 4. Why?

14.  Line 272-282: the numbers do not match the data presented in Fig 6. More precise description is required.

15.  Line 289-292: references are required for the statement.

16.  Discussion about the temporal evolution in the seropositive participants across different district regions should be included.

17.  Line 360: the description is not precise because the mean value of IC50 against Beta variant (Table 3) is the lowest which is not the newest variant.

18.  Line 374: reference is required.

19.  It is highly recommended to compare the seroprevalence and the mortality rate in Ticino, which could be another strong evidence to support the importance of launching vaccine campaign in the future.

20.  Proof-reading is recommended to be done carefully. There are several grammatical errors.

Author Response

(The authors gave the same response as above.)

Round 2

Reviewer 3 Report

The revised version is highly improved, but a number of errors or concerns need to be further addressed.

Major comments:

1.       The resolution of Fig 1 needs to be further improved.

2.       To have a clear presentation, the following response should be included in the Method session:

“anti-N antibodies wane more quickly than anti-S antibodies. If a participant only had anti-S antibodies, but reported not to have been vaccinated, his/her antibodies had been developed following an infection, irrespective of the negativity of the anti-Nucleocapsid antibodies.”

3.       In Fig 4, it is recommended to change the y-axis label to be “Percentage of candidates (%)”.

4.       The legend of Fig 6 should be presented after the figure. It is recommended to change the y-axis label to be “Percentage of candidates with antibody activity higher than IC50 (%)”. Moreover, the y-axis number should be changed to “100, 80,…” but not “1, 0.8,….”.

5.       Line 357-358: it is recommended to quote Fig 3.

Author Response

We thank this Reviewer for the suggestions provided in this second round of revision. Please see the attachment for the point-by-point response to the comments.
